# Microstructural Adaptation for Prey Manipulation in the Millipede Assassin Bugs (Hemiptera: Reduviidae: Ectrichodiinae)

**DOI:** 10.3390/biology12101299

**Published:** 2023-09-30

**Authors:** Shiyu Zha, Zhiyao Wang, Xinyu Li, Zhaoyang Chen, Jianyun Wang, Hu Li, Wanzhi Cai, Li Tian

**Affiliations:** 1Department of Entomology and MOA Key Lab of Pest Monitoring and Green Management, College of Plant Protection, China Agricultural University, Beijing 100193, China; 2020319010224@cau.edu.cn (S.Z.); 2020319010430@cau.edu.cn (Z.W.); lixinyubjfu@bjfu.edu.cn (X.L.); zhaoyangchen@cau.edu.cn (Z.C.); tigerleecau@hotmail.com (H.L.); caiwz@cau.edu.cn (W.C.); 2College of Forestry, Beijing Forestry University, Qinghua East Road 35, Beijing 100083, China; 3Environment and Plant Protection Institute, Chinese Academy of Tropical Agricultural Sciences, Haikou 571101, China; wjycau@gmail.com

**Keywords:** Reduviidae, stenophagy, antennae, mouthparts, sensillum, microstructures

## Abstract

**Simple Summary:**

The millipede assassin bugs are well known for their prey specialization. In this study, we examined the fine structures of the antennae, mouthparts, and legs of four genera and species of ectrichodiines using a scanning electronic microscope and compared them with those in Tribelocephalinae, a lineage closely related to Ectrichodiinae. We found that ectrichodiines possess distinctly more slightly transverse ridges on the mandibles, which probably facilitate stabilizing mandibular stylets in prey tissues. The small papillae on the legs are considered to adapt to immobilize millipedes. Overall, our study illustrates, at a microstructural level, the remarkable morphological adaption of prey manipulation in ectrichodiines, which will facilitate future studies on the adaptive evolution of feeding habits in Reduviidae.

**Abstract:**

Species in Ectrichodiinae are known for their prey specialization on millipedes. However, knowledge of the morphological adaptations to this unique feeding habit was limited. In the current study, we examined the microstructures of the antennae, mouthparts, and legs of four millipede feeding ectrichodiines, *Ectrychotes andreae* (Thunberg, 1888), *Haematoloecha limbata* Miller, 1953, *Labidocoris pectoralis* (Stål, 1863), and *Neozirta eidmanni* (Taueber, 1930), and compared them with those of three species of tribelocephalines, a group closely related to Ectrichodiinae. On the antennae, we found four types of antennal sensilla. On the mouthparts, we recognized four types of labial sensilla. Sampled ectrichodiines have distinctly more and denser slightly transverse ridges on the external side of mandibles than tribelocephalines. *E. andreae* and *H. limbata* possess numerous small papillae fringed with densely arranged finger-print-like grains on the trochanter and femur; these probably facilitate the immobilization of prey. Overall, our study illustrates, at a microstructural level, the remarkable morphological adaption of prey manipulation in ectrichodiine, and has enhanced our understanding about stenophagy in the family Reduviidae.

## 1. Introduction

The millipedes (Diplopoda) are a diverse group of arthropods that are known to produce noxious chemical secretions as defenses [1,2], making them distasteful to many carnivores, but some animals have evolved to be specialized predators of millipedes and can avoid and/or tolerate their chemical defenses. Immobilizing the millipede prey as quickly as possible to prevent it from producing enough defensive secretions was suggested to be one of the mechanisms for the predator to overcome the chemical defenses. For instance, the phengodid beetle larvae *Phengodes laticollis* Kutschera, 1864 bites the millipede to inject toxin through its mandibles, and the millipede is paralyzed almost immediately and unable to open the valves of its repugnatorial glands to release defensive secretions [1,3,4]. Another group of specialized predators of millipedes that paralyze the preys by injecting toxin are the millipede assassin bugs (Hemiptera: Reduviidae: Ectrichodiinae). Ectrichodiinae is a diverse clade with more than 700 known species, and at least 14 species in 13 genera of this group have been observed to feed on millipedes [5,6].

Preying on millipedes poses potential challenges for these assassin bugs. First, detecting prey-derived chemical cues to locate prey from a long distance requires the assistance of olfactory sensory organs [7,8,9]. Second, millipedes are usually armed with chemical defenses and hard, smooth body walls, and are often much larger than these assassin bugs, which makes them difficult to be manipulated [5,6,9,10]. Previous studies have demonstrated several behavioral adaptation and morphological innovations which help millipede assassin bugs to cope with the challenges of prey localization and manipulation. For instance, Haridass and Ananthakrishnan observed feeding responses of *Haematorrhophus nigroviolaceus* (Reuter, 1881) to different artificial baits offered in the laboratory, and showed that this ectrichodiine may mainly rely on vision to locate its millipede prey in short-distance with potential auxiliary of olfaction [10]. A prior sporadic study reported the hook-shaped ultimate segment of the labium and the large spatulate apex with many slightly transverse ridges on the mandibles of ectrichodiines that may have adapted to millipede feeding [6]. The legs of many reduviids also evolved diverse morphological modifications and adaptations for prey capture, including hairy attachment structures known as the “fossula spongiosa”, chelate forelegs, and elaborate armature [11,12,13,14]. These leg structures are suggested to be necessary to grasp prey tightly before the toxic saliva immobilizes it. The fossula spongiosa is a cushion-shaped expanded area ventrodistally on the tibia consisting of tenent hairs with spatulate or tapering apices interspersed with sensory setae, which are widespread among Reduviidae [11]. In addition to fossula spongiosa, some ectrichodiines have numerous small papillae on the trochanters and femora, which has been considered as a taxonomic character within Ectrichodiinae in previous studies [14,15]. The potential biological function of these small papillae requires observations of their microstructure. However, study on the morphology of these mentioned structures is largely undocumented in Ectrichodiinae, and more comparative analyses of these are needed to understand their biological function.

In this study, we examined the fine structure of the antennae, mouthparts, and legs (Figure 1) of four millipede feeding species in four genera of Ectrichodiinae, and compared them with closely related species in Tribelocephalinae. Based on the comparative study, we identify structural features that possibility involved in adaptation to millipede feeding. The study will expand our knowledge of the mechanisms underlying stenophagy in Reduviidae as well as other predatory insect lineages.

## 2. Materials and Methods

### 2.1. Insect Collection

The samples comprise seven species in this study, including *Ectrychotes andreae* (Thunberg, 1888), *Haematoloecha limbata* Miller, 1953, *Labidocoris pectoralis* (Stål, 1863), and *Neozirta eidmanni* (Taueber, 1930) in Ectrichodiinae [16], and *Opistoplatys majusculas* Distant, 1940 [17], an unidentified *Opistoplatys* species (*Opistoplatys* sp.), and *Tribelocephala limbata* China, 1940 [18] in Tribelocephalinae (Hemiptera: Reduviidae). Tribelocephalinae were considered to be phylogenetically related to Ectrichodiinae, and these two subfamilies were synonymized in previous studies [15]. Members of Tribelocephalinae have been reported to feed on ants, termites, and blattids [19], which allows them to be used as alternative congeners for morphological comparisons with Ectrichodiinae to understand the adaptations of the latter to millipede feeding [20]. Adults of *E*. *andreae*, *H*. *limbata*, and *L.*
*pectoralis* were collected in Jiufeng Park, Beijing, China, in July–August 2022. *N. eidmanni* were collected in Dayaoshan, Guangxi, China, in May 2022. *O. majusculas* were collected in Huilonghe, Yunnan, China, in November 2021. *O.* sp. were collected in Jindong Grand Canyon, Beijing, China. *T. walkeri* were collected in Xishuangbanna, Yunnan, China. Samples for SEM were preserved in 75% ethanol and stored at 4 °C. Three males and three females of each Ectrichodiinae species, *O. majusculas*, and *T. walkeri* were prepared for morphological examination. A male of *O.* sp. was examined due to the extremely small number of available specimens. Additional adults of species of Ectrichodiinae were collected to observe the feeding behavior on millipedes.

### 2.2. Sample Preparation for SEM

Samples for SEM were prepared following methodology previously described by Wang et al., 2020 [6], with minor adjustments. In general, antennae, labia, stylet bundles, and legs of the examined specimens were removed from the bodies under an OPTEC (Chongqing, China) stereoscope, and were cleaned three times using an ultrasonic cleaner (KQ2200E, Kunshan Ultrasonic Instrument Co, Ltd, Kunshan, Jiangsu, China.) for 30 s each time. The samples were then dehydrated through serial baths of 85%, 90%, and 95% each for 15 min, and 100% ethanol twice each for 10 min, followed by drying in air. The prepared samples were then coated with a film of gold (EIKO IB-2), and then imaged with a scanning electron microscope HITACHI S-3400N at 10 kV in the scanning microscopy laboratories of the College of Biological Science of China Agricultural University. The morphological terminology follows Cobben [21] and Schneider [22], and more specialized nomenclature from other studies was used in addition [6,23,24,25].

### 2.3. Feeding Behavior Observing

Millipede assassin bugs were reared in a transparent plastic box 190 mm long, 125 mm wide, and 75 mm high. The bottom of the box was covered with cotton wool about 10 mm high, which was moistened with water. Several pieces of egg cartons were placed on the cotton wool to provide these insects with a place of shelter and rest, similar to the natural environment. The feeding conditions were a temperature of 26 ± 0.5 °C and RH of 70 ± 10%. Some common Diplopoda species in China, including *Spirobolus bungii* Brandt, 1833 [26] in Spirobolida (spirostreptid millipedes), *Orthomorphella pekuensis* (Karsch, 1881) [27], *Helicorthomorpha holstii* (Pocock, 1895) [28], and *Oxidus gracilis* (Koch, 1847) [29] in Polydesmida (polydesmid millipedes) were collected as prey. Adult individuals of Ectrichodiinae were offered with millipedes after 72 h of fasting. The predatory behavior was captured using a Canon EOS 7D camera for later analysis.

## 3. Results

### 3.1. Type and Distribution of Antennal Sensilla

The antennae of sampled species comprise three segments: a scape, a subdivided pedicel, and a subdivided distiflagellomere (Figure 1A). Each antennal segment is approximately cylindrical except that the distiflagellomere is slightly narrowed at the apex (Figure 2A,B,F). The scape and pedicel are similar in diameter, while the basiflagellomere and distiflagellomere are significantly thinner than them. Four types of antennal sensilla were found in this study, including sensilla trichodea, sensilla basiconica, sensilla chaetica, and trichobothria (Table 1).

Antennal sensilla trichodea (AnTr) are hair-shaped structures with tapered tips. Three subtypes of AnTr can be distinguished among the sampled species. Antennal sensilla trichodea I (AnTr I) are long and broad, possessing flexible sockets (Figure 2 and Figure 3A,D,G,I,K). Antennal sensilla trichodea II (AnTr II) are shorter and slender than AnTr I (Figure 3B,E,J,K,L) and have inflexible sockets. The socket of AnTr II of *N. eidmanni* is located in a cavity with raised edges (Figure 3J). Antennal sensilla trichodea III (AnTr III) with flexible sockets are the shortest trichodea (Figure 3C,F,H). In all sampled species, AnTr are located on all the antennal segments.

Antennal sensilla basiconica (AnBa) are peg-shaped with blunt tips, and are inserted into inflexible sockets (Figure 4A–E). Approximately two-thirds of the wall of AnBa was covered with longitudinal grooves, leaving the remaining one-thirds relatively smooth. AnBa can be found only on basiflagellomere and distiflagellomere in all the surveyed species.

Antennal sensilla chaetica (AnCh) are straight and stiff, with deep grooves on the walls. This type has sharp tips and flexible sockets (Figure 4F–J). In all sampled species, AnCh are located on the pedicel, basiflagellomere, and distiflagellomere.

Antennal trichobothria (AnTb) are long and very flexible (Figure 5). The sockets of AnTb are set within a circular depression (cd) surrounded by an oval membrane (om). In the six examined species, AnTb are significantly rarer in number and only distribute on the pedicel.

### 3.2. Type and Distribution of Labial Sensilla

Numerous sensilla are laterally arranged on the labial apex, consisting of sensilla trichodea, sensilla campaniformia, sensilla basiconica, and sensilla placoidea (Table 2).

Labial sensilla trichodea (LaTr) are hair-shaped with a sharp tip, mainly situated at the margin of the labial apex. Two subtypes of LaTr can be distinguished by size and shape in this study. Labial sensilla trichodea I (LaTr I) are long, curved, and have flexible sockets (Figure 6A and Figure 7E). Labial sensilla trichodea II (LaTr II) are short and straight, with a smooth surface and inflexible sockets (Figure 6A,C–E, Figure 7A,B and Figure 8A,D). These were found in all samples in this study.

Labial sensilla campaniformia (LaCa) are oval, dome-shaped structures, mainly distributed on the ventral surface of the labial tip (Figure 6B,F and Figure 8D,F). These are present in all sampled Ectrichodiinae species.

Labial sensilla basiconica (LaBa) are peg-shaped, and possess smooth walls. Three subtypes of LaBa were distinguished. Labial sensilla basiconica I (LaBa I) have slightly narrowed tips and inflexible sockets (Figure 6A,C,F, Figure 7D and Figure 8A–C,E). Labial sensilla basiconica II (LaBa II) are short and blunt-tipped, and embedded in inflexible sockets surrounded by a swollen cuticle (Figure 6B,D,E, Figure 7A,B and Figure 8C,F). Labial sensilla basiconica III (LaBa III) are morphologically similar to LaBa II but significantly smaller and shorter, and have inflexible sockets (Figure 8F). LaBa I and LaBa II occur in both Ectrichodiinae and Tribelocephalinae species, while LaBa III were only found in *N. eidmanni*.

A single pair of labial sensilla placoidea (LaPl) were detected in all examined samples. These are elongated, dome-shaped, and are located on the lateral sides of the labial tip (Figure 6E, Figure 7A,C and Figure 8A,D,E).

### 3.3. Fine Structure of Mandibles and Maxillae

Stylet fascicle is comprised of two separate mandibular stylets (Md) (Figure 9) and two interlocked maxillary stylets (Mx) (Figure 10 and Figure 11). Mandibular stylets are attached to, and surround, maxillary stylets which are slightly longer than them.

The left and right mandibles are mirror-symmetrical in sampled species. The apex of mandibles is flat, oval, and spatulate, with a slightly narrowed end (Figure 9). Many slightly transverse ridges (str) are located on the external side of each mandibular stylet, and the shape and number vary among species. These ridges are faint, small, and mainly scale-shaped in *E. andreae* (Figure 9B) while they are line-shaped in the other sampled species (Figure 9E,I,L). In the examined ectrichodiine species, slightly transverse ridges are distinctly more numerous and denser than those of tribelocephalines. *Haematoloecha limbata* have about 290 slightly transverse ridges on their mandibles, which is the largest number among the species measured, and these values are 220 in *L. pectoralis*, 190 in *N. eidmanni*, 90 in *O. majusculas*, and 70 in *T. walkeri*. Many longitudinal ridges (lr) extend to the base of the spatulate apex (Figure 9N,J). Numerous longitudinal ridges dorsally and ventrally distribute on the inner surface of the spatulate apex (Figure 9C,F), and *L. pectoralis* possess some small projections (sp) arranged longitudinally in a row near the ventral longitudinal ridges (Figure 9F).

In the seven surveyed species, the maxillary stylets are left–right asymmetrical and interlocked, forming a food canal (Fc) and salivary canal (Sc) (Figure 10A,D and Figure 11B–D,F). The left maxillary stylet (LMx) has a straight, sharp-tipped apex with many small teeth (sto). Ectrichodiinae species possess a small narrow lobe (nlo) on the apex of the left maxillary stylet (Figure 10B and Figure 11C). On the internal surface of the apex of the left maxillary stylet, there are several small spines (ssp) (Figure 10C and Figure 11C). There are some transverse ridges (tr) and short bristles (sbr) on the subapical region of the inner side of the left maxilla in ectrichodiines (Figure 10A,C,D,E and Figure 11C,D). The short bristles are absent on the inner side of the left maxillary stylet in sampled tribelocephalines. Many small pores (spo) distribute on the external surface of the left maxillary stylet (Figure 10B). The apex of the right maxillary stylet (RMx) is tapered, and slightly curved with a blunt tip (Figure 10F and Figure 11B,D). There are ventral (vr) and dorsal rows (dr) of curved hair-shaped short bristles (sbr) on the internal side of each right maxillary stylet, and the ventral rows possess many lamellate structures (lss) (Figure 11A,D).

### 3.4. Small Papillae on Legs

The leg of the sampled species consists of a coxa, a trochanter, a femur, a tibia, and a tarsus (Figure 1B–D). Small papillae (spp) were documented on the ventral side of legs in both males and females in *E. andreae* and *H. limbata*. The small papillae are present on the trochanters and femora of the fore- and midlegs in two species, and *E. andreae* possess these on the hind-trochanter as well (Figure 12A–C,E,F). In *E. andreae*, small papillae are close to each other forming a cluster on the trochanter, while they are arranged in a strip on the femur. In *H. limbata*, the small papillae were sparsely distributed over the ventral side of the trochanter and the basal part of the femur. The small papillae are egg-shaped in *E. andreae* while hemispherical in *H. limbata*. These papillae were covered with dense, finger-print-like grains (flg) (Figure 12D,G).

### 3.5. Predatory Behavior of Ectrichodiinae Species

Millipede predation by sampled millipede assassin bugs generally involves eight steps: arousing, locating, approaching, paralyzing, resting, sucking, releasing, and cleaning. After being aroused by nearby prey, the predator will turn its head and sway the antennae to locate the prey. This is followed by approaching and eventually catching the prey. During prey manipulation, the predator will be immobilize its millipede prey using the forelegs and midlegs, during which the fossula spongiosa is always attached to the surface of the prey, which is against the trochanter and the base of the femur (Figure 13A,B,D,E). After the prey is immobilized, the predator will insert its stylets into the site of the millipede, usually through the intersegmental membranes along the ventral and/or ventro-lateral trunk (Figure 13) to inject toxic saliva to paralyze the prey. After the prey is paralyzed, the predator will clean its body and antennae with forelegs, rub the ventral body surfaces on the ground, and drag the paralyzed prey to a safe place for consumption. The feeding process may last for hours. In addition, *H. limbata* accepted all millipede species as prey, while *N. eidmanni* rejected any polydesmid millipedes and *E. andreae*, *L. pectoralis* rejected *S. bungii* even after fasting.

## 4. Discussion

Stenophagy (as opposed to euryphagy) is an ecological strategy with a narrow trophic niche and it particularly occurs in many herbivorous and parasitic animal taxa, and occasionally in carnivores [30]. Stenophagous predators often possess specialized morphological traits which facilitate the utilization of their exclusive prey [31]. The Reduviidae (also known as assassin bugs) are one of the most species-rich predatory insect groups, which comprises more than 6600 species [32]. A number of species in this group are stenophagous insects, specializing in a particular type of prey, such as termites, ants, spiders, and millipedes [14,32]. Some species also sport morphological and behavioral adaptations for prey manipulation [33,34,35]. The presently provided data identify a series of morphological structures associated with the stenophagous habit of millipede assassin bugs.

### 4.1. Morphological Adaptations of the Mouthparts for Millipede Feeding

Adaptive shifts of feeding habits and feeding-related organs have been hypothesized to be a main driver of true bug radiation [25,36]. The mandibular stylets not only protect and stabilize the maxillary stylets, but also play an important role in penetrating the tissues of the host/prey [21,37]. The mandibles of heteropterans are known to undergo morphological modifications to be adapted to various diet types. For instance, the prominent and stout teeth on the mandibular stylets of some pyrrhocorid species were considered to facilitate breaking hard seed coats [23,38]; the mandibular teeth of predatory pentatomids are sharp and hook-shaped, while they are short and blunt in plant feeders [25,39,40]; the blood-feeding triatomine species possess a single row of hook-shaped teeth on each mandible, used to penetrate the skin of vertebrate hosts [41]. In the current study, we have demonstrated that all examined species in Ectrichodiinae have distinctly more and denser transverse ridges on the external sides of mandibles than those in tribelocephalines. These densely arranged ridges may facilitate stabilizing and dragging large and vigorously moving prey like millipedes. A similar observation and hypothesis were also made by a previously study on *Haematoloecha nigrorufa* (Stål, 1867) [6]. Prey preferences for Tribelocephalinae are largely unclear, and this group was recorded as appearing to be generalists in the literature we searched [19], suggesting that a greater number of transverse ridges on the external sides of mandibles is associated with millipede feeding in the clade Ectrichodiinae + Tribelocephalinae. More records on the diet of tribelocephalines are required to understand the evolution of feeding habits in this clade. In the genus *Haematoloecha*, *H. limbata* have more slightly transverse ridges (about 290) than previously reported in *H. nigrorufa* (about 150). According to our laboratory and field observation, we found that *H. limbata* accepted both polydesmid millipedes and larger spirostreptid millipedes as prey (Figure 13B,E), while *H. nigrorufa* rejected spirostreptid millipedes even after fasting, only feeding on polydesmid millipedes (personal observations). Similarly, only polydesmid millipedes were photographed to be preyed on by *H. nigrorufa* in the previous study [6]. These further imply the association between increased mandibular ridges and large-sized millipede prey. More field observations and behavioral experiments are necessary to provide statistically significant data to investigate the feeding preferences of millipede species within Ectrichodiinae. In sampled species, *E. andreae* have scale-shaped slightly transverse ridges (str), making their number incomparable with those line-shaped ridges in other species, which suggests that the shape of slightly transverse ridges on the external side of mandibles could serve as a taxonomic characteristic within Ectrichodiinae. Rather faint transverse ridges on the outer stylet surface were considered as a synapomorphy of both Ectrichodiinae and Tribelocephalinae, and the mandibular stylets in most Reduviidae are beset with between ten and a maximum of 35 transverse ridges [12]. The lamellate structures in the internal ventral side of the right maxillary stylet are another feature observed in Ectrichodiinae and Tribelocephalinae, and their function is not yet clear [6,12].

### 4.2. Morphological Adaptation of the Legs for Millipede Feeding

Previous studies have demonstrated various structures on the legs of assassin bugs (usually the forelegs), which potentially function to facilitate prey capture [14,42]. For example, forelegs are modified into chelate or subchelas in Phymatinae (ambush bugs) [43]. Some Harpactorinae species have a sticky substance coating on the forelegs [13], and pronounced spines on the forelegs were observed in Emesinae (thread-legged bugs) and some Reduviinae species [14,44]. Fine structures of these typical raptorial legs usually facilitate gripping relatively small-to-medium-sized prey. In contrast, millipede assassin bugs need to cope with preys that are usually much larger than themselves and have a smooth and hard body surface. Thus, holding and stabilizing prey would be a challenge for them during predation [6,10]. Many Ectrichodiinae species have small papillae on their trochanters and femora, making their modified legs indicative of raptorial legs [14]. In our study, we have shown, for the first time, the fine structure of these small papillae. They are numerous cuticular appendages, and are egg-shaped in *E. andreae*, while hemispherical in *H. limbata*. Numerous finger-print-like grains are densely arranged on the small papilla. These grains may serve to increase the roughness of the contact surface with the body of millipede prey. During predation, the cushion-shaped fossula spongiosa ventrodistally located on the tibia can attach to the millipede, and the gripping action can make the prey’s body surface press tightly against these small papillae on the trochanter and femur as observed, thereby producing considerable friction so that the prey can be immobilized. Accordingly, we suggest that small papillae can be regarded as auxiliary structures of fossula spongiosa, and that the morphological separation of these two allows the millipede assassin bug to adjust the angle between the femur and tibia to adapt prey of different sizes. Cuticular structures with a similar morphology were also noted on the edge of the fore femur in Gelastocoridae, and were hypothesized to be useful for better grasping prey [38]. Not all ectrichodiines possess the small papillae (e.g., *L. pectoralis* and *N. eidmanni* in the present study), and, in some species, there are sexual dimorphisms determining whether these structures are present (e.g., *Racelda spurca* (Stål, 1860)*, Gibbosella planiscutum* Forthman, Chlond & Weirauch, 2016, and *Toxopus griswoldi* Forthman, Chlond & Weirauch, 2016) [15]. More empirical evidence is needed to validate the function of the small papillae.

### 4.3. Sensillar System on Antennae and Labium

Antennal sensilla of insects were evidenced to have the function of receiving chemical and mechanical signals from their surroundings, playing important roles in host and mate location [24,25,45,46,47,48,49,50]. Antennal sensilla of ectrichodiines have been reported for the first time in this study. Three types of antennal sensilla were distinguished by morphology and distribution. Sensilla trichodea are the dominant sensilla on antennae in sampled reduviid species, and these could be divided into four morphological distinct subtypes (AnTr I, II, III) [25,46]. Antennal trichobothria with circular depression and an oval membrane are relatively scarce and present only on pedicel. This type was suggested to respond to air movements in previously published studies [51,52]. Antennal sensilla chaetica have grooves and flexible sockets, and this type was categorized as typical mechanosensilla, which have functions of detection and transmission of various mechanical stimuli [46]. Sensilla basiconica, with deep longitudinal grooves and an inflexible socket, probably perceive chemical stimuli regarding the host location and sexual recognition [24,25,46,53]. Millipede assassin bugs had been observed swaying antennae toward their prey during predation [6], which may represent a signal-receiving behavior. Previous research has demonstrated that the chemical stimuli from living preys could enhance the feeding responses of millipede assassin bugs [10]. The scent of defensive secretions released by millipede after being attacked was speculated to attract other individuals of millipede assassin bugs, usually nymphs, and lead to communal predation [5,21].

Labial sensilla of heteropterans are known to play an important function in host detection [25,38,40,54,55]. Adapted to stridulating by rubbing the tip of the labium against the transversely striate prosternal sulcus, sensilla on the labial tip in Reduviidae are relatively shorter and flatter than those in other terrestrial true bugs [12,56,57]. Labial sensilla of our surveyed millipede bugs are morphologically similar to those of *H. nigrorufa* and other reduviids [6,56]. Four types of labial sensilla were distinguished. Labial sensilla trichodea with flexible sockets are considered to be mechanoreceptive and may play a particular role in determining the appropriate site of prey for inserting the stylets [25,56]. Mechanoreceptors may also include dome-shaped sensilla campaniformia, which were usually observed located in various places susceptible to the tension and deformation of many insect species, and were shown to possess a proprioreceptory function in Peiratinae [56]. Sensilla basiconica on the labial tip were considered chemoreceptive structures and probably act as gustatory receptors [6,56]. The pores on the labial sensilla basiconica of *H. nigrorufa* support its chemoreceptor function [6]. Sensilla placoidea are also considered chemoreceptors, and most true bugs have a single pair on the labial tip [6,56,57,58,59].

## 5. Conclusions

The fine-structural characterization of antennae, mouthparts, and small papillae on the legs of some ectrichodiines is provided in this study. The basal pattern of the antennal sensilla and labial sensilla in sampled species does not obviously differ from other reduviids. The slightly transverse ridges (str) on the external side of mandibles in Ectrichodiinae are significantly more numerous than those in Tribelocephalinae. The small papillae with densely finger-print-like grains (flg) were observed in *E. andreae* and *H. limbata*. These mentioned structures in millipede assassin bugs may represent morphological adaptations associated with the manipulation of their exclusive preys.

## Figures and Tables

**Figure 1 biology-12-01299-f001:**
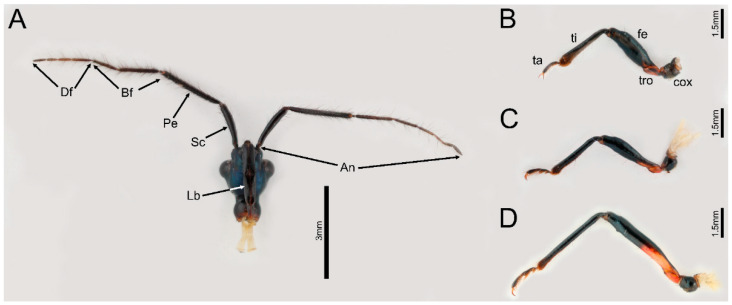
General morphology of head and legs of *Ectrychotes andreae* (**A**) head, (**B**) foreleg, (**C**) midleg, (**D**) hindleg. Abbreviations: An, antennae; Bf, basiflagellomere; cox, coxa; Df, distiflagellomere; fe, femur; Lb, labium; Pe, pedicel; Sc, scape; ta, tarsus; ti, tibia; tro, trochanter.

**Figure 2 biology-12-01299-f002:**
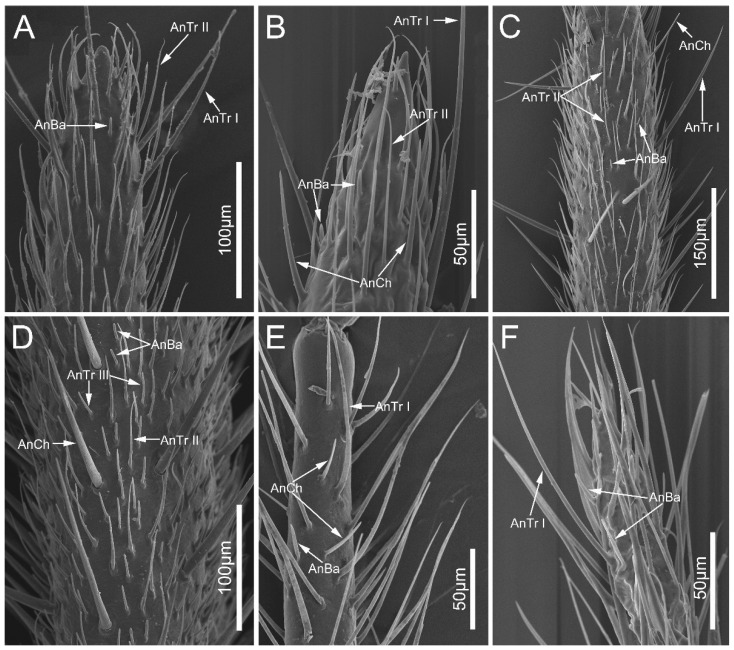
The antennal flagellum of the sampled reduviid species, showing the distribution of antennal sensilla trichodea, sensilla chaetica, and sensilla basiconica. (**A**) *Ectrychotes andreae*, (**B**) *Haematoloecha limbata*, (**C**) *Labidocoris pectoralis*, (**D**) *Neozirta eidmanni*, (**E**) *Opistoplatys majusculas*, and (**F**) *Opistoplatys* sp. Abbreviations: AnBa, antennal sensilla basiconica; AnCh, antennal sensilla chaeticaand; AnTr I–III, antennal sensilla trichodea I–III.

**Figure 3 biology-12-01299-f003:**
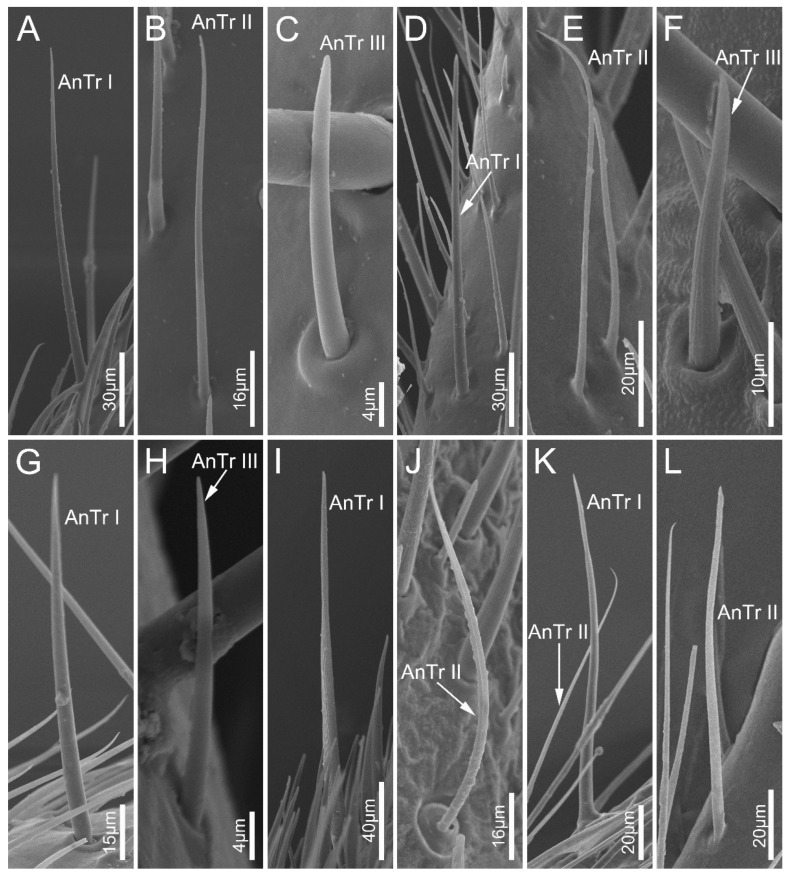
Antennal sensilla trichodea on the antennal flagellum of the sampled reduviid species. (**A**–**C**) *Ectrychotes andreae*; (**D**–**F**) *Haematoloecha limbata*; (**G**–**I**) *Labidocoris pectoralis*; (**J**–**L**) *Neozirta eidmanni*. AnTr I–III, antennal sensilla trichodea I–III.

**Figure 4 biology-12-01299-f004:**
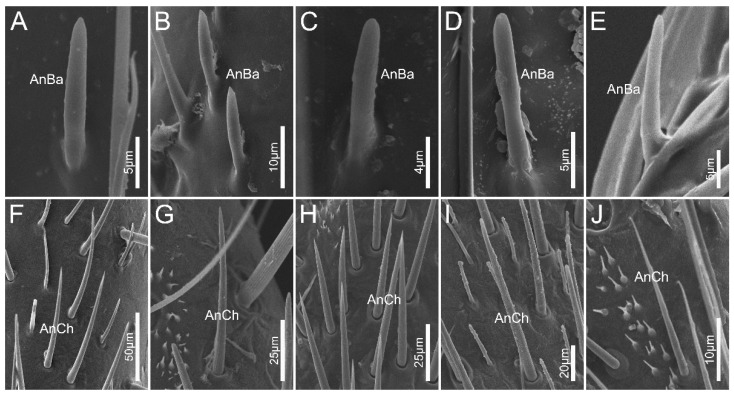
Antennal sensilla sensilla basiconica and antennal sensilla trichodea IV of the sampled reduviid species. (**A**,**F**) *Ectrychotes andreae*; (**B**,**G**) *Haematoloecha limbata*; (**C**,**H**) *Labidocoris pectoralis*; (**D**,**I**) *Neozirta eidmanni*; (**E**,**J**) *Opistoplatys majusculas*. Abbreviations: AnBa, antennal sensilla basiconica; AnCh, antennal sensilla chaetica.

**Figure 5 biology-12-01299-f005:**
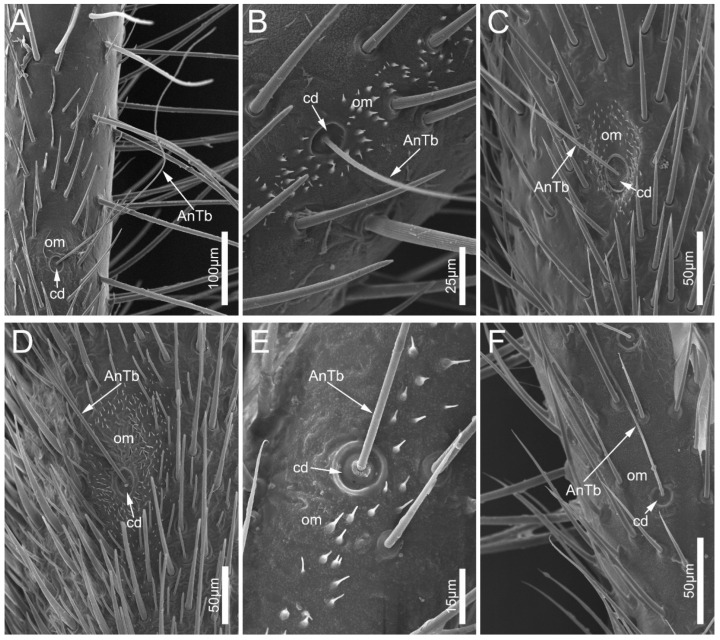
Antennal trichobothria on the antennal pedicel of the sampled reduviid species. (**A**) *Ectrychotes andreae*; (**B**) *Haematoloecha limbata*; (**C**) *Labidocoris pectoralis*; (**D**) *Neozirta eidmanni*; (**E**) *Opistoplatys majusculas*; and (**F**) *Opistoplatys* sp. Abbreviations: AnTb, antennal trichobothria; cd, circular depression; and om, oval membrane.

**Figure 6 biology-12-01299-f006:**
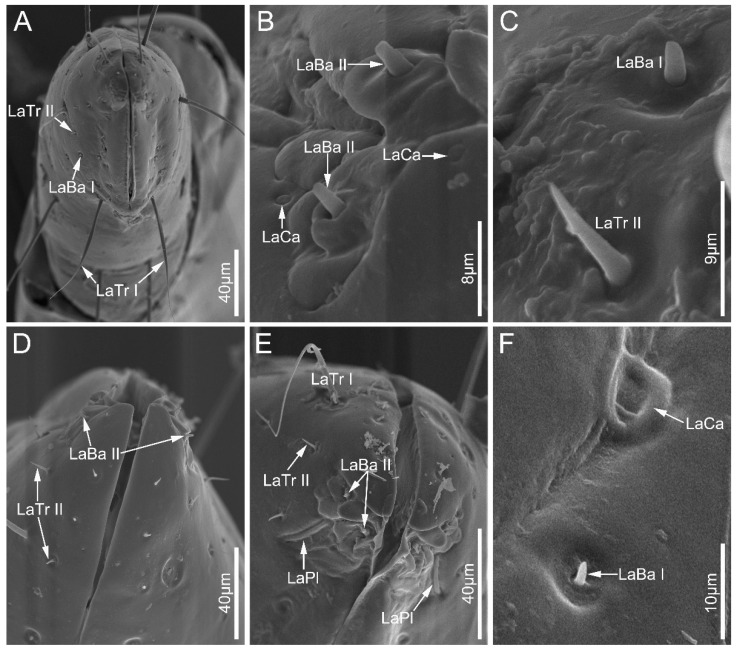
Sensilla on labial apex of *Ectrychotes andreae* (**A**–**C**) and *Haematoloecha walkeri* (**D**–**F**). Abbreviations: LaBa I–II, labial sensilla basiconica I–II; LaCa, labial sensilla campaniformia; LaTr I–II, labial sensilla trichodea I–II; and LaPl, sensilla placoidea.

**Figure 7 biology-12-01299-f007:**
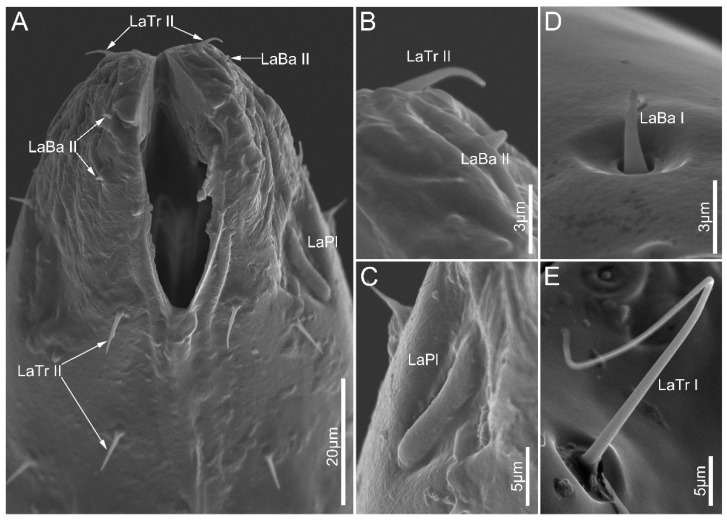
Sensilla on labial apex of *Tribelocephala walkeri* (**A**–**C**), *Opistoplatys majusculas* (**D**), and *Opistoplatys* sp.(**E**). Abbreviations: LaBa I–II, labial sensilla basiconica I–II; LaTr I–II, labial sensilla trichodea I–II; and LaPl, sensilla placoidea.

**Figure 8 biology-12-01299-f008:**
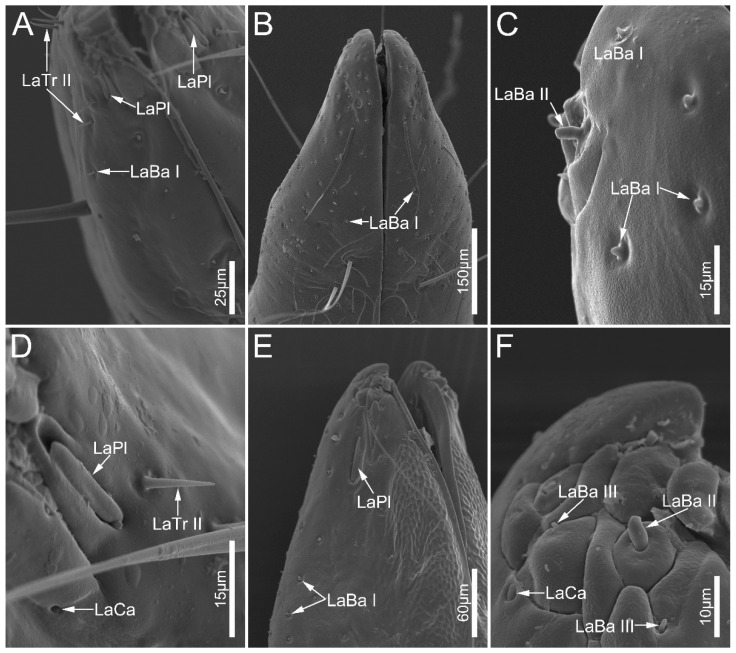
Sensilla on labial apex of *Labidocoris pectoralis* (**A**,**D**) and *Neozirta eidmanni* (**B**,**C**,**E**,**F**). Abbreviations: LaBa I–III, labial sensilla basiconica I–III; LaCa, labial sensilla campaniformia; LaTr II, labial sensilla trichodea II; and LaPl, sensilla placoidea.

**Figure 9 biology-12-01299-f009:**
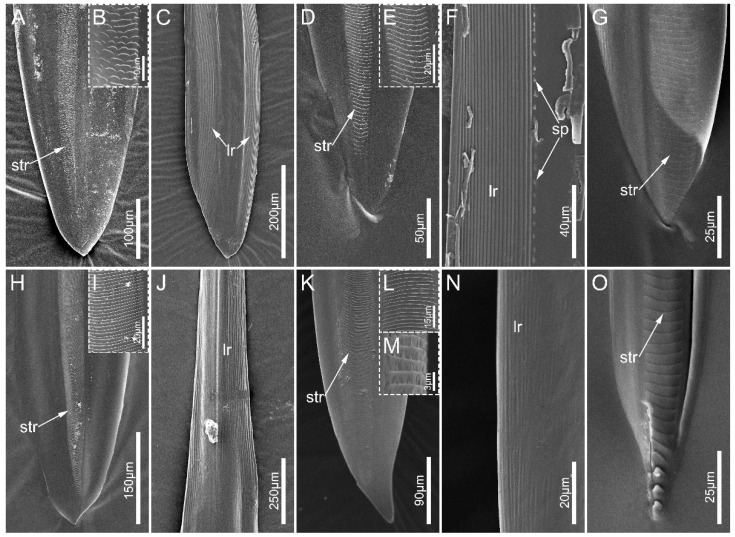
Mandibular stylets of the sampled reduviid species. (**A**–**C**) *Ectrychotes andreae*; (**D**–**F**) *Labidocoris pectoralis*; (**G**) *Opistoplatys majusculas*; (**H**–**J**) *Haematoloecha limbata*; (**K**–**N**) *Neozirta eidmanni*; (**O**) *Tribelocephala walkeri*. (**B**,**E**,**I**,**L**) showing enlarged view of slightly transverse ridges; (**M**) showing the lateral details of slightly transverse ridges. Abbreviations: sp. small projections; str, slightly transverse ridges; lr, longitudinal ridges.

**Figure 10 biology-12-01299-f010:**
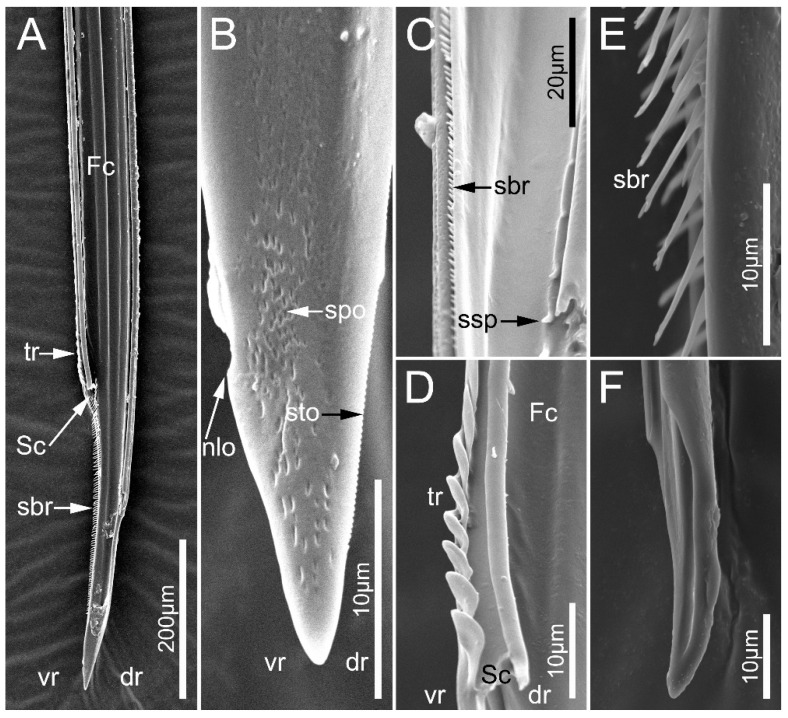
Maxillary stylets of the sampled reduviid species. (**A**,**B**) *Ectrychotes andreae*; (**C**,**D**) *Haematoloecha limbata*; (**E**,**F**) *Labidocoris pectoralis*. (**A**–**D**) left maxillae; (**E**,**F**) right maxillae. Abbreviations: dr, dorsal row; Fc, food canal; nlo, narrow lobe; sbr, short bristles; Sc, salivary; spo, small pore; ssp, short spines; sto, small tooth; tr, transverse ridges; and vr, ventral row.

**Figure 11 biology-12-01299-f011:**
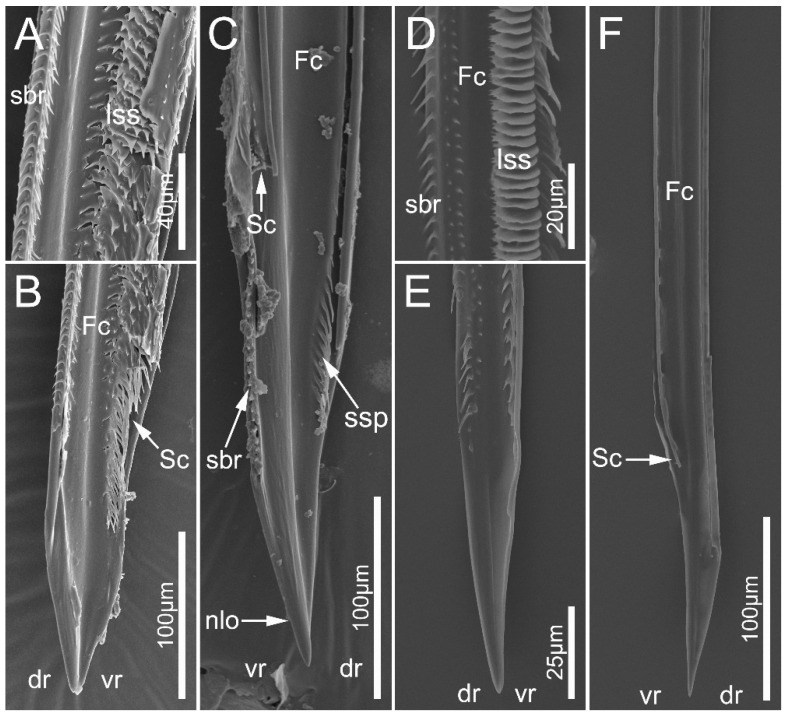
Maxillary stylets of the sampled reduviid species. (**A**–**C**) *Neozirta eidmanni*; (**D**–**F**) *Tribelocephala walkeri*. (**C**,**F**) left maxillae; (**A**,**B**,**D**,**E**) right maxillae. Abbreviations: dr, dorsal row; Fc, food canal; lss, lamellate-shaped structures; nlo, narrow lobe; sbr, short bristles; Sc, salivary; ssp, short spines; and vr, ventral row.

**Figure 12 biology-12-01299-f012:**
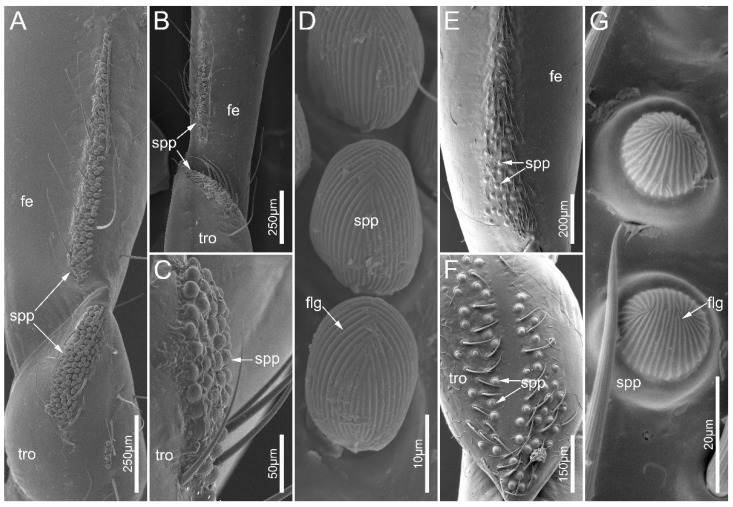
Small papillae on legs of *Ectrychotes andreae* and *Haematoloecha limbata*. (**A**–**D**) *Ectrychotes andreae*; (**E**–**G**) *Haematoloecha limbata*. (**A**,**E**,**F**) forelegs; (**B**) midlegs; and (**C**) hindlegs. (**D**,**G**) showing enlarged view of the small papillae. Abbreviations: fe, femur; flg, finger-print-like grains; spp, small papillae; tro, trochanter.

**Figure 13 biology-12-01299-f013:**
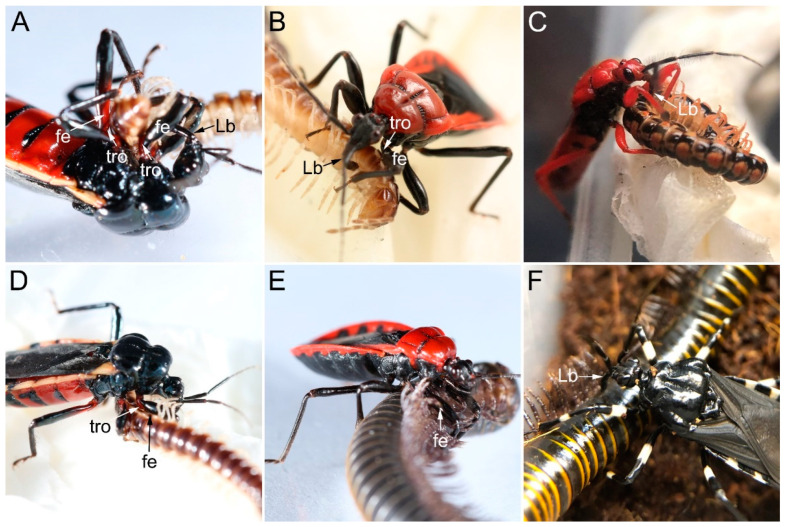
Preying behavior of sampled ectrichodiines on millipede. (**A**,**D**) *Ectrychotes andreae* preying on *Oxidus gracilis*; (**B**) *Haematoloecha limbata* preying on *Oxidus gracilis*; (**C**) *Labidocoris pectoralis* preying on *Helicorthomorpha holstii*; (**E**) *Haematoloecha limbata* preying on *Spirobolus bungii*; (**F**) *Neozirta eidmanni* preying on *Spirobolus bungii*. Abbreviations: fe, femur; Lb, labium; and tro, trochanter.

**Table 1 biology-12-01299-t001:** Characteristics of the identified types of antennal sensilla.

Type	Localization	Shape	Tip	Sockets
AnBa	flagellum	peg-shaped	blunt	inflexible
AnCh	pedicel, basiflagellomere, distiflagellomere	straight and stiff	sharp	flexible
AnTr I	pedicel, basiflagellomere, distiflagellomere	hair-shaped	tapered and sharp	inflexible
AnTr II	pedicel, basiflagellomere, distiflagellomere	hair-shaped	sharp	inflexible
AnTr III	all antennal segments	hair-shaped	sharp	inflexible
AnTb	pedicel	hair-shaped	sharp	set in a circular depression

AnBa, antennal sensilla basiconica; AnCh, antennal sensilla chaetica; AnTb, antennal trichobothria; and AnTr I-III, antennal sensilla trichodea I–III.

**Table 2 biology-12-01299-t002:** Characteristics of the identified types of labial sensilla.

Type	Localization	Shape	Sockets	Tip	Observed in Species
LaBa I	labial tip	peg-shaped	inflexible	blunt	all sampled species
LaBa II	labial tip	peg-shaped	inflexible	blunt	all sampled species
LaBa III	labial tip	peg-shaped	inflexible	blunt	*Neozirta eidmanni*
LaCa	ventral surface of labial tip	oval and dome-shaped	/	/	all sampled ectrichodiinae species
LaPl	lateral sides of the labial tip	dome-shaped	/	/	all sampled species
LaTr I	margin of labial apex	hair-shaped	flexible	sharp	all sampled species
LaTr II	labial apex	hair-shaped	inflexible	sharp	all sampled species

LaBa I–III, labial sensilla basiconica I–III; LaCa, labial sensilla campaniformia; LaTr I–II, labial sensilla trichodea I–II; and LaPl, sensilla placoidea.

## Data Availability

All data generated or analyzed during this study are included in this published article.

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
