# Peer review of "Microstructural Adaptation for Prey Manipulation in the Millipede Assassin Bugs (Hemiptera: Reduviidae: Ectrichodiinae)"

_biology, 2023, doi:10.3390/biology12101299_

Round 1
Reviewer 1 Report
Dear Authors,
With great interest, I undertook the review of the manuscript "Microstructural Adaptation for Prey Manipulation in the Millipede Assassin Bugs (Hemiptera: Reduviidae: Ectrichodiinae)". Unfortunately, the manuscript turned out to be disappointing. I believe it needs significant improvement, including additional research. Therefore, now I do not make minor remarks on the editing of the text, but only a few points that authors should consider thoroughly.
1. The manuscript lacks a proper taxonomy regime. All species names should be listed at least once in complete form (with the author of the description and the year), and the relevant source items (descriptive works) should be quoted.
2. It would be helpful to include a photograph or a micrograph:
a) the entire tentacle of the selected species
b) the entire labium of the selected species
c) side view of the front leg of the selected species.
This would certainly give an idea of ​​the subject of the study.
3. What is the actual difference between sensilla trichoidea, s, chaetica and s. basiconica? The distinction seems discretionary and is not based on subjective characteristics (shape).
4. The authors write:
"AnTr IV with circular depression and oval membrane are relatively scarce and present only on the pedicel. This subtype was described as "trichobothria" in previously published studies and suggested to respond to air movements [44]".
Why was the term trichobothria abandoned? Certainly, the described structures are trichobothria. I encourage you to study a few papers on this type of structure:
Hemala, V., Kment, P., Malenovský, I., 2020. The comparative morphology of adult pregenital abdominal ventrites and trichobothria in Pyrrhocoroidea (Hemiptera: Heteroptera: Pentatomomorpha). Zool. Anz. 284, 88–117. https://doi.org/10.1016/j.jcz.2019.11.006.
Gao, C.-Q., Rédei, D., Shi, X.-Q., Cai, B., Liang, K., Gao, S., Bu, W.-J., 2017. A comparative study of the abdominal trichobothria of Trichophora, with emphasis on Lygaeoidea (Hemiptera: Heteroptera). Eur. J. Entomol. 114, 587–602. https://doi.org/10.14411/eje.2017.072.
Taszakowski A., Gorczyca J., Herczek A. 2020. Comparative study of the cephalic trichobothria in plant bugs (Hemiptera: Heteroptera: Miridae). Micron 137, 102918: 1-10. doi: 10.1016/j.micron.2020.102918
The authors write:
"Antennal sensilla trichodea IV (AnTr IV) are long and crooked (Figure 3)."
This is not their feature. On the contrary, they are simple but very flexible. Therefore, when observed in the SEM, they appear to be bent.
5. SEM micrographs allow to assess only the shape and arrangement of the antennal and labial sensilla. Their quality gives practically no information about the surface. Do they have pores? This essential feature allows you to determine the function of individual sensilla. Detailed analysis of the surface, especially the antennal sensilla, probably allows for finding more types of sensory structures. The adherent sensilla likely have pores, indicating chemoreceptor function. I recommend reading the following paper:
Taszakowski A., Masłowski A., Daane K.M., Brożek J. 2023. Closer view of antennal sensory organs of two Leptoglossus species (Insecta, Hemiptera, Coreidae). Scientific Reports 13 (617). https://doi.org/10.1038/s41598-023-27837-4
6. Why have only "small papillae" been studied for legs? The legs, especially the first pair, should be subjected to a comprehensive analysis. It would also be advisable to take a cursory look at the legs of the 2nd and 3rd pair, which can also be used in feeding.
Overall, the manuscript is an exciting topic, but it definitely needs some improvement, especially regarding the sensory organs on the antennae.
Best wishes
Author Response
All replies are included in the file submitted.

Reviewer 2 Report
I have marked all comments in the manuscript.

Author Response

(The authors gave the same response as above.)

Reviewer 3 Report
Abstract: I would like to inquire about the reason for the omission of years following the descriptors.
In section 2.2, concerning Sample Preparation for SEM, it is crucial to establish a connection with the detailed flowchart of Scanning Electron Microscopy (SEM) stages. I wish to clarify whether any reference source was adopted for this procedure or if this proposition constitutes a novel innovation. It is imperative that this premise be adequately elucidated.
Regarding Figure 3E, I am encountering difficulties in discerning the circular depression labeled as CD. Kindly conduct a verification to ensure the accuracy of this element.
Author Response

(The authors gave the same response as above.)

Round 2
Reviewer 1 Report
Dear Authors,
Below are two minor notes.
- Please consider the correctness of the description of Figure 3J. The socket of the sensillum shown in this figure appears to be different from that in Figures 3E and 3L.
- Please add a blank line after the description of Figure 4.
Best wishes
Author Response
Please see the attachment. All replies are included in the file submitted.
